# p166 links membrane and intramitochondrial modules of the trypanosomal tripartite attachment complex

**Bernd Schimanski**[1]*, **Salome Aeschlimann**[1,2], **Philip Stettler**[1,2], **Sandro Käser**[1], **Maria Gomez-Fabra Gala**[3,4,5], **Julian Bender**[6], **Bettina Warscheid**[6], **F.-Nora Vögtle**[3,7¤], **André Schneider**[1]*

**1** Department of Chemistry, Biochemistry and Pharmaceutical Sciences, University of Bern, Bern, Switzerland, **2** Graduate School for Cellular and Biomedical Sciences, University of Bern, Bern, Switzerland, **3** Institute of Biochemistry and Molecular Biology, ZBMZ, Faculty of Medicine, University of Freiburg, Freiburg, Germany, **4** Faculty of Biology, University of Freiburg, Freiburg, Germany, **5** Spemann Graduate School of Biology and Medicine, University of Freiburg, Freiburg, Germany, **6** Biochemistry II, Theodor Boveri-Institute, Biocenter, University of Würzburg, Würzburg, Germany, **7** CIBSS—Centre for Integrative Biological Signalling Studies, University of Freiburg, Freiburg, Germany

¤ Current address: Center for Molecular Biology of Heidelberg University (ZMBH), DKFZ-ZMBH Alliance and Network Aging Research, University of Heidelberg, Heidelberg, Germany
* bernd.schimanski@unibe.ch (BS); andre.schneider@unibe.ch (AS)

**Data Availability Statement:** All relevant data are within the manuscript and its Supporting Information files, The mass spectrometry

## Abstract

The protist parasite *Trypanosoma brucei* has a single mitochondrion with a single unit genome termed kinetoplast DNA (kDNA). Faithfull segregation of replicated kDNA is ensured by a complicated structure termed tripartite attachment complex (TAC). The TAC physically links the basal body of the flagellum with the kDNA spanning the two mitochondrial membranes. Here, we characterized p166 as the only known TAC subunit that is anchored in the inner membrane. Its C-terminal transmembrane domain separates the protein into a large N-terminal region that interacts with the kDNA-localized TAC102 and a 34 aa C-tail that binds to the intermembrane space-exposed loop of the integral outer membrane protein TAC60. Whereas the outer membrane region requires four essential subunits for proper TAC function, the inner membrane integral p166, via its interaction with TAC60 and TAC102, would theoretically suffice to bridge the distance between the OM and the kDNA. Surprisingly, non-functional p166 lacking the C-terminal 34 aa still localizes to the TAC region. This suggests the existence of additional TAC-associated proteins which loosely bind to non-functional p166 lacking the C-terminal 34 aa and keep it at the TAC. However, binding of full length p166 to these TAC-associated proteins alone would not be sufficient to withstand the mechanical load imposed by the segregating basal bodies.

## Author summary

Mitochondria evolved from a single endosymbiotic event and are a hallmark of eukaryotes. The large majority of genes for mitochondrial proteins are nuclear encoded now and only a small number are found in the mitochondrial genome. The protist *Trypanosoma*

proteomics data have been deposited to the ProteomeXchange Consortium via the PRIDE partner repository with the dataset identifier PXD033042. http://proteomecentral. proteomexchange.org/cgi/GetDataset?ID= PXD033042.

**Funding:** This work was supported by the following grants: NCCR RNA & Disease, a National Centre of Competence in Research, supported by the Swiss National Science Foundation, grant number 182880, to AS Schweizerischer Nationalfonds zur Förderung der Wissenschaftlichen Forschung, grant number 175563, to AS Deutsche Forschungsgemeinschaft, grant number 390939984, to FNV SFB1381, grant number 403222702, to FNV Emmy-Noether Programm, to FNV Deutsche Forschungsgemeinschaft, grant number, 403222702/SFB 1381, to BW The funders had no role in study design, data collection and analysis, decision to publish, or preparation of the manuscript.

**Competing interests:** The authors have declared that no competing interests exist.

*brucei* is an extreme eukaryote in many aspects. For instance, trypanosomes have a single mitochondrion and its genome–called kinetoplast DNA (kDNA)–locates as a single unit inside the mitochondrion close to the basal body of the flagellum. The tripartite attachment complex (TAC) forms a connection between the basal body and the kDNA ensuring faithful segregation of kDNA among the daughter cells upon cytokinesis. Recently, several TAC subunits of the cytoplasm, the outer mitochondrial membrane (OM) and the mitochondrial matrix have been characterized. Here, we identify p166 as the first TAC subunit of the inner mitochondrial membrane. It is anchored with a single transmembrane domain separating the protein into a N-terminal moiety located in the matrix and a short C-tail. The latter reaches into the intermembrane space and binds the OM subunit TAC60 whereas the N-terminus interacts with the matrix subunit TAC102. Thus, with p166 we identified the missing link required to connect different modules of the TAC.

## Introduction

Mitochondria have an own genome which reflects their bacterial ancestry. Moreover, even though most mitochondrial proteins are imported from the cytosol, the few organellar-encoded proteins are essential for mitochondrial function [1–3]. Eukaryotes therefore require mechanisms to guarantee that the mitochondria and their genomes are faithfully divided and segregated to the daughter cells during cell division [4–6]. The parasitic protozoon *Trypanosoma brucei* is an extreme example for this. Unlike yeast and mammals, which have numerous mitochondria each having multiple genomes, it has only a single mitochondrion with a single unit genome, termed kinetoplast DNA (kDNA), that is localized to a specific region within the organelle [7, 8]. The kDNA consists of two highly intercalated populations of circular DNA molecules that form a disc-like structure composed of a few dozens of maxicircles (22 kb in length) and a few thousand of minicircles (1 kb in length) [9, 10]. Maxicircles are functionally analogous to the mitochondrial genome of other species. They encode 18 protein-coding genes many of which are cryptogenes, meaning their transcripts need to be extensively edited to convert them into functional mRNAs. The minicircles are heterogenous in sequence and encode guide RNAs serving as templates to mediate RNA editing [11–13]. Unlike suggested for yeast and mammals whose mitochondrial genomes are distributed within the organelle, stochastic partition cannot be the mechanism to segregate the old and the newly replicated kDNAs to the daughter cells. Instead, trypanosomes have a hardwired structure, termed tripartite attachment complex (TAC), that connects the matrix located kDNA disc across the two mitochondrial membranes with the basal body of the flagellum and thus couples segregation of the replicated kDNA to the segregation of the old and the new flagellum in dividing cells [10, 14]. The TAC is restricted to kinetoplastids. However, a recent study suggests that the concept of coupling inheritance of organelles to flagellar segregation also applies to mitosomes of the anaerobic protist *Giardia*. In these cells a subgroup of mitosomes is linked to axonemes of the oldest flagella [15, 16] raising the question whether this arrangement is an ancestral or a derived trait.

Segregation of the trypanosomal kDNA works as follows. During replication of the kDNA, which is tightly coordinated with cell cycle and occurs shortly before the onset of the S-phase, the kDNA network doubles in size [7, 8]. Subsequently a new TAC is formed that links the new basal body to the replicated kDNA disc. Finally, segregation of the old and the new flagellum will separate the old and the new kDNA networks, before the mitochondrion is divided in two between the pair of basal bodies and the kDNA discs during cytokinesis [17].

The TAC can be divided into three morphologically defined regions: (i) the exclusion zone filaments (EZF) that connect the basal body to the mitochondrial outer membrane (OM); (ii) the differentiated membranes (DM) where OM and inner membrane (IM) are closely opposed and (iii) the unilateral filaments (ULF) inside mitochondria which connect the IM to the kDNA disc [18]. In their pioneering study Zhao et al. discovered the first TAC subunit, named p166, by screening an RNAi library for kDNA loss [19]. Since then seven further essential TAC subunits have been characterized that are localized to all three TAC regions [14].

TAC assembly is a complex problem. It has been shown that the three TAC regions are formed in a hierarchical and temporally controlled way, starting at the basal body with the EZF, which after their formation connect to the TAC subunits in the DM region [20]. Finally, the ULF are formed which link the IM to the newly replicated kDNA disc. TAC biogenesis therefore requires the coordinated assembly of a cytosolic module (EZF), a module consisting of integral mitochondrial OM and IM proteins (DM) and an intramitochondrial module (ULF) [20]. The DM module is of special interest because it links the extramitochondrial EZF tether with the intramitochondrial ULF tether. The OM part of this module is amazingly complex. It consists of four essential integral membrane TAC subunits: the beta-barrel proteins TAC40 [21] and TAC42, TAC60 [22], and pATOM36 [23], which besides being a TAC subunit is also involved in the biogenesis of OM proteins [23, 24]. All of these subunits are inserted into the OM with the help of the atypical protein translocase of OM (ATOM) that is localized all over the surface of the mitochondrion [25]. After membrane insertion the OM TAC subunits are transported to the small very defined region of the DM between the basal body and the kDNA, in what is arguably the most extreme lateral sorting event known for any mitochondrion. In contrast to the OM not a single integral membrane TAC subunit has been identified in the IM. p166 would be an interesting candidate for an IM TAC subunit, as it contains a predicted N-terminal mitochondrial targeting sequence (MTS) and a transmembrane domain (TMD) near its C-terminus. However, a variant of p166 which lacks the TMD still correctly localized to the TAC, indicating that its localization does not depend on the TMD [19].

Here we have investigated the function and the biogenesis of p166. We show that the protein is indeed the missing integral IM TAC subunit. Furthermore, we demonstrate that its C-terminus is oriented towards the intermembrane space (IMS) where it interacts with the IMS-exposed loop of the OM TAC subunit TAC60. Finally, our data indicate that assembly of p166 into the TAC is a two-step process involving low affinity interactions with so far unknown TAC-associated proteins and a high affinity with OM TAC subunits to stably lock it in the growing TAC structure.

## Results

### p166 is an essential TAC component of the mitochondrial IM

In a first step to characterize p166 we established a cell line capable of RNAi-mediated downregulation using the 3' UTR of the p166 mRNA as target (Fig 1A, left panel). Knockdown of p166 leads to a growth retardation around 3 days after induction of RNAi. A quantitative analysis of kDNAs by fluorescent microscopy 2 days after induction of RNAi revealed a remarkable loss of the mitochondrial genome in more than 75% of cells. As a hallmark of failed kDNA segregation cells with smaller (15%) and enlarged overreplicated kDNAs (6%) could be observed (Fig 1B, left panel). Overall, these data agree well with previous findings about p166 [19] and all other described TAC components [14]. In a next step the RNAi cell line was further transfected with a construct leading to the integration of a C-terminal triple HA tag sequence in one allele of p166 resulting in the expression of an *in situ* epitope-tagged version of p166 (p166-HA, Fig 2A). Importantly, the tagged gene acquired with the tubulin intergenic region a

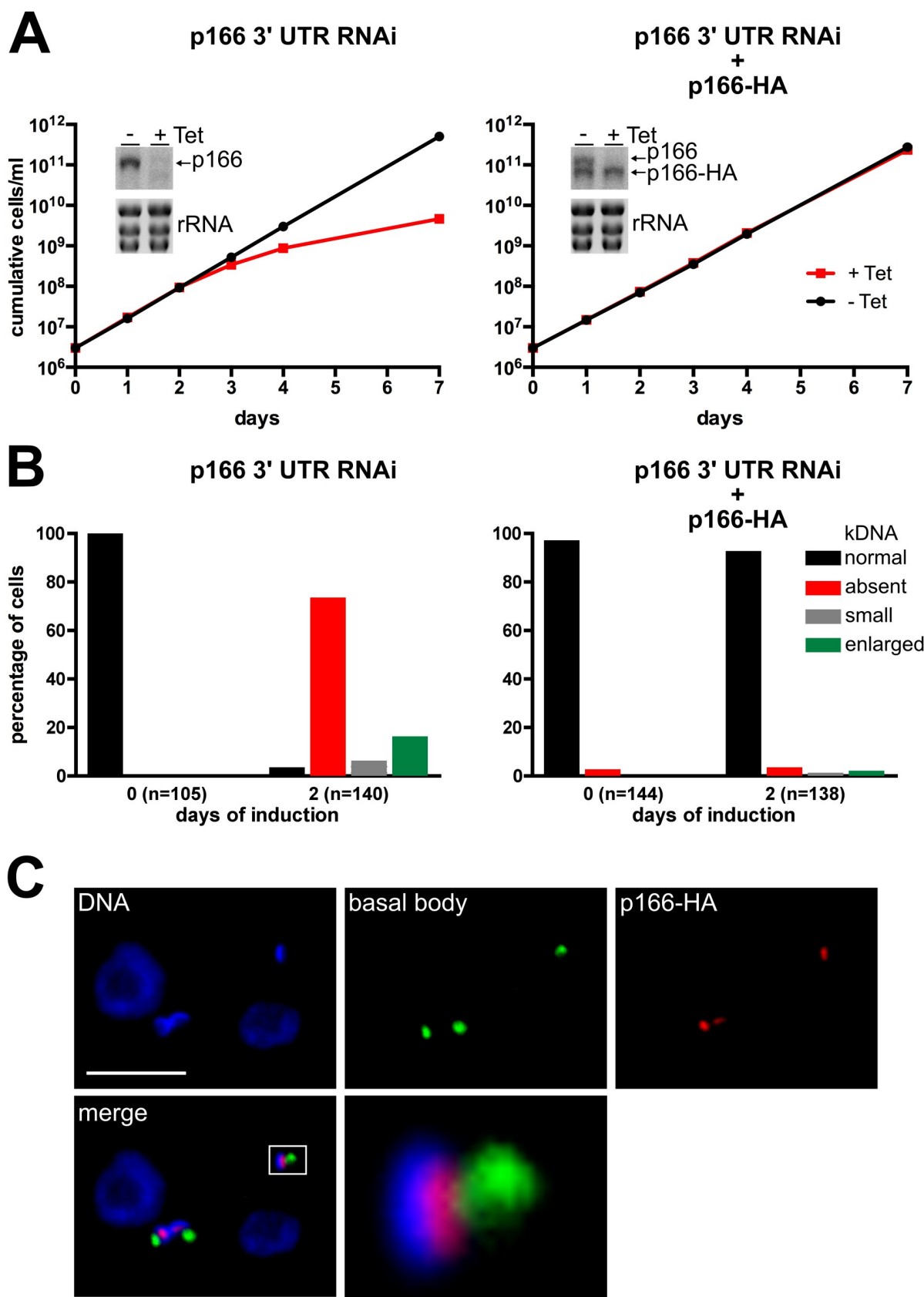

**Fig 1. In situ tagged p166-HA is functional. (A)** Growth analysis of cells induced for RNAi of p166 in absence (left) or presence (right) of a RNAi resistant allele encoding *in situ* tagged p166-HA. In both growth curves inserts showing a northern blot analysis are included to demonstrate effects of tetracyclin (Tet) induced RNAi against the indicated mRNAs after 2 days of induction. Ethidiumbromide stained ribosomal RNA (rRNA) bands serve as control for equal loading. **(B)** DNA of uninduced (left) and RNAi induced cells (right) was stained with DAPI and kDNA content and phenotype were analyzed by fluorescence microscopy in the indicated number of individual cells. **(C)** Immunofluorescene microscopy with whole cells. DNA was stained with DAPI, the basal body with anti YL1/2 and tagged p166 was detected with anti-HA. The enlarged part of the merged picture is indicated by a white rectangle. The scale bar shows 5 μm.

different 3' UTR and is therefore resistant to RNAi and constitutively expressed in the absence or presence of wild-type p166 (Fig 1A, right panel). Analysis of cell growth indicates that the expression of *in situ* tagged p166-HA can fully complement for the loss of wild-type p166 upon RNAi (Fig 1A, right panel). Furthermore, no changes in kDNA levels can be observed in cells expressing both wild-type and tagged p166 and in cells expressing the tagged version only (Fig 1B, right panel). Finally, the subcellular localization of the tagged p166 was investigated by immunofluorescence microscopy using uninduced whole cells as samples (Fig 1C). The result shows that p166-HA localizes between the basal body of the flagellum and the kDNA which is in line with previous findings [19]. We therefore conclude that tagged p166 is correctly localized, that the tag does not interfere with p166 function and that all further results using p166-HA are representative for wild-type p166 as well.

p166 consists of 1501 amino acids, contains a predicted MTS at the N-terminus and a single predicted TMD close to the C-terminus (Fig 2A). Biochemical analysis confirmed that p166 is indeed found in the mitochondria-enriched pellet after digitonin fractionation together with the mitochondrial OM protein ATOM40 and in contrast to the cytoplasmic protein EF1a. Furthermore, after alkaline carbonate extraction of mitochondria p166 is found in the pellet fraction together with ATOM40 indicating that p166 is an integral membrane protein (Fig 2B) [26]. To exclude that fractionation of p166 into the carbonate pellet is an artefact due to aggregation, the pellet was treated with Triton-X-100 which showed that, as expected for an integral membrane protein, p166 is solubilized (S1A Fig). We also analyzed a HA-tagged version of p166 that lacks the TMD, termed p166-ΔTMD-HA. Intriguingly, while still fractionating with the organellar fraction p166-ΔTMD was exclusively recovered in the supernatant of the carbonate extraction (S1B Fig). This indicates that the predicted C-terminal TMD is both necessary and sufficient for membrane integration of p166.

In a previous study the subcellular localization of known TAC components including p166 was analyzed and their distance to the basal body was measured [20]. p166 was shown to be more distal to the basal body than the OM TAC components TAC60 and TAC40 and closer to the matrix subunit TAC102. This indicates that membrane integral p166 is located in the IM of mitochondria. Furthermore, the predicted presence of a MTS indicates that p166 is indeed a IM protein. To test if the predicted MTS sequence is functioning as a mitochondrial import signal *in vivo*, trypanosomes were transfected with different constructs allowing for inducible expression of GFP. As shown in Fig 2C GFP localizes to the cytoplasm. In contrast, GFP versions containing the previously characterized 15 amino acids long MTS of mitochondrial heat shock protein 60 (mtHSP60) or the first 25 amino acids of p166 at their N-termini are efficiently imported into the mitochondrion. In line with the immunofluorescence results (Fig 2C) the p166 MTS GFP fusion protein is recovered in the organellar pellet in a digitonin extraction (Fig 2D). Intriguingly, a slower migrating band of low intensity is also detected in whole cells but is absent from the organellar pellet. Translocation of the MTS across the inner membrane depends on the membrane potential and its disruption abolishes import of MTS-containing proteins across and into the IM. Fig 2E shows that the intensity of the slower migrating band in total cellular extracts increased in a time-dependent manner after addition

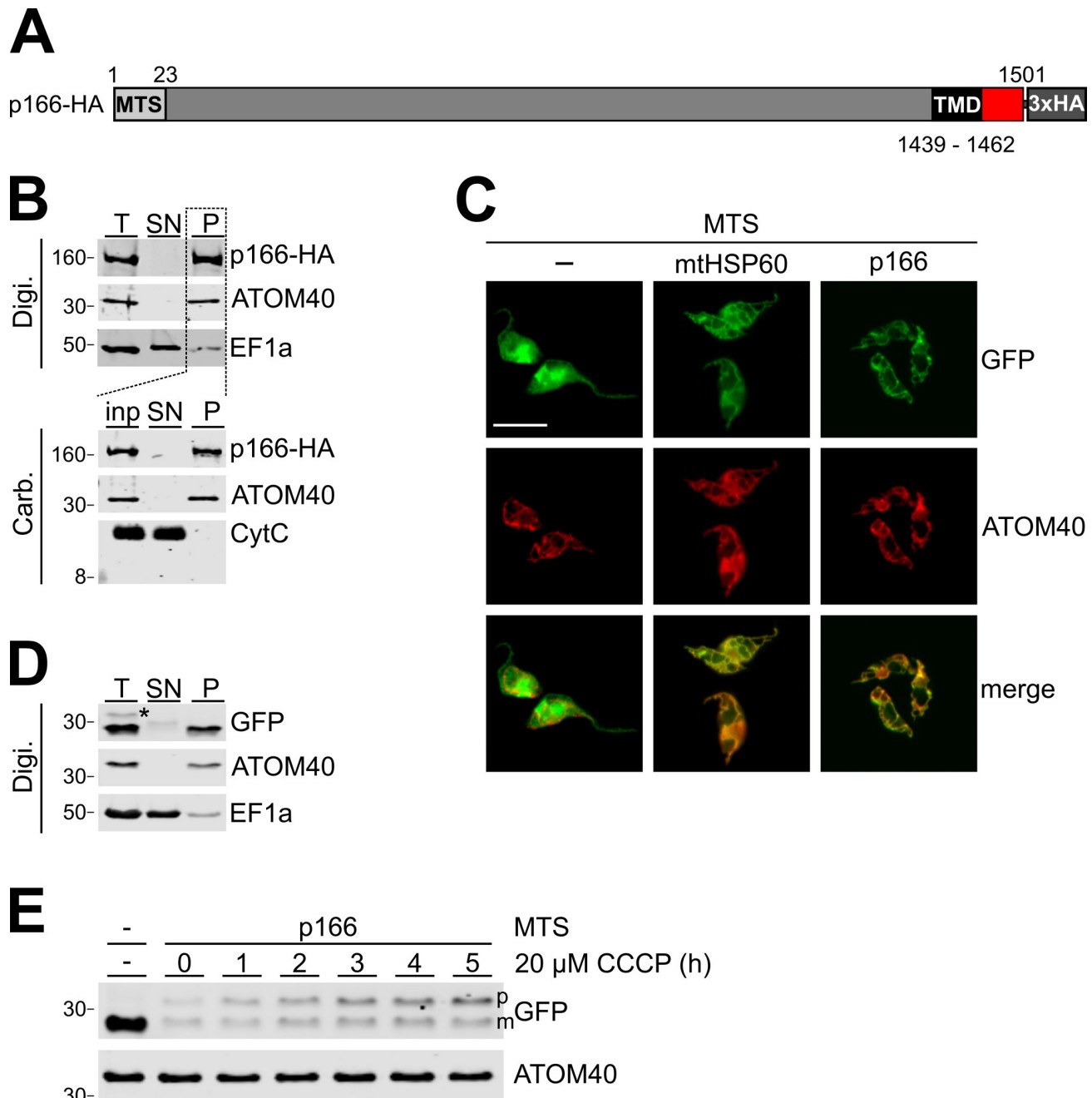

**Fig 2. p166-HA is an integral membrane protein of the IM. (A)** Schematic depiction (not to scale) of *in situ* tagged p166. A predicted mitochondrial targeting sequence (MTS) and transmembrane domain (TMD) are indicated. Numbers represent the position of amino acid residues. **(B)** Biochemical fractionation of cell extracts. For digitonin (Digi.) extraction same cell equivalents of total cells (T), supernatant (SN) and pellet (P) were separated by SDS-PAGE and p166-HA, the mitochondrial outer membrane protein ATOM40 and the cytoplasmic marker EF1a were detected by immunoblot. The pellet served further as input (inp.) for alkaline carbonate (Carb.) extraction and was separated into supernatant and a final pellet. On immunoblots p166-HA, ATOM40 and the soluble protein cytochrome C (CytC) were detected. Numbers on the left indicate sizes of molecular weight marker bands in kDa. **(C)** Fluorescence microscopy analysis of tetracycline inducible cells expressing variants of GFP with indicated mitochondrial targeting sequences (MTS) fused to the N-terminus. Staining for ATOM40 serves as marker for mitochondria. **(D)** Digitonin extraction as in (B) of the cell line expressing GFP with the p166 MTS. Asterisk indicates the putative unprocessed precursor protein. **(E)** The cell line expressing GFP with the p166 MTS was induced for 2 hours and subsequently treated with the uncoupler CCCP as indicated. Subsequently total cellular extract was analyzed by SDS-PAGE. Left most lane shows GFP alone. Position of precursor p166-MTS-GFP (p) and mature GFP are indicated (m). ATOM40 serves as a loading control.

of the protonophore carbonyl cyanide m-chlorophenyl hydrazine (CCCP). This indicates that in the absence of the membrane potential p166 MTS GFP is not imported anymore resulting in cytosolic accumulation of uncleaved precursor protein. Thus, we conclude that p166 is an integral IM protein whose import depends on the membrane potential-dependent translocation of its MTS across the IM before it is processed in the matrix.

## The C-tail of p166 is indispensable for function

The previous study by Zhao et al. [19] indicated that the C-tail of p166 is not necessary for correct localization. However, these experiments were performed in cells expressing both wild-type and tagged C-terminally truncated p166 simultaneously. It was therefore not possible to determine if the C-tail is needed for the proper function of p166. To address this question we modified one allele of p166 in the cell line capable of RNAi against the p166 3'UTR with a construct leading to the expression of a RNAi resistant *in situ* tagged version of p166 lacking the last 34 C-terminal amino acids (p166-ΔC-HA, Fig 3A). Digitonin fractionation and alkaline carbonate extraction confirmed that p166-ΔC-HA still behaves like an integral mitochondrial membrane protein (Fig 3B). However, cell growth analysis after RNAi-mediated ablation of wild-type p166 revealed that p166-ΔC-HA is not able to complement for the loss of endogenous p166 (Fig 3C). The observed growth phenotype is due to a pronounced kDNA segregation defect with the majority of cells losing their kDNA after 2 days of induction. Again and similar to downregulation of p166 alone (Fig 1B), more than 70% of the cells analyzed show no detectable kDNA whereas around 7% and 17% of the cells harbour small and enlarged kDNAs, respectively (Fig 3D). These aberrant kDNAs and the localization of the tagged p166 variant can be visualized by immunofluorescene microscopy (Fig 3E, middle and right panel). The results show that the C-terminal truncated p166 is correctly localized between the basal bodies and the small and overreplicated kDNAs. Furthermore, even in cells which lack kDNA p166-ΔC-HA is still associated with the basal bodies, indicating that the correct localization of p166 is independent of the presence of kDNA (Fig 3E, right panel).

## The C-tail of p166 reaches into the IMS

The results presented so far strongly indicate that the C-terminus of p166 is essential for proper TAC function. In a next step we wanted to investigate the topology of p166. The fact that in the DM region of the TAC the OM and IM are closely opposed and therefore essentially exclude any IMS strongly suggests that the large soluble 150 kDa N-terminal part of p166 is oriented towards the matrix side.

   Full length variants of p166 that are functional are firmly integrated into the TAC and therefore insoluble (S2 Fig) which makes them difficult to study. Thus, we established two individual cell lines expressing mini-versions of p166 that lack the N-terminal 1359 amino acids (mini-p166-HA) as well as in the second case the 34 amino acid C-tail distal to the TMD (mini-p166-ΔC-HA) (Fig 4A). Since the deleted N-terminal domain of p166 also contains the MTS, the 15 amino acids long MTS of mtHSP60 was fused to the N-termini of both variants. As a result, the truncated fusion proteins are imported into the mitochondrion and can be found in the membrane fraction (S3 Fig). Next, the cell growth and kDNA segregation were monitored upon expression of the truncated proteins. A slight slow growth phenotype starting at day 4 after induction could be observed in cells expressing mini-p166-HA with an intact C-terminus (Fig 4B, left panel). Moreover, analysis of the kDNA content after 2 days of induction revealed a striking decrease of cells containing normal kDNA and a simultaneous appearance of cells with small or overreplicated kDNAs. At day 8 after induction this effect was even more pronounced (Fig 4C, left panel). In contrast, cells expressing mini-p166-ΔC-HA, which lacks

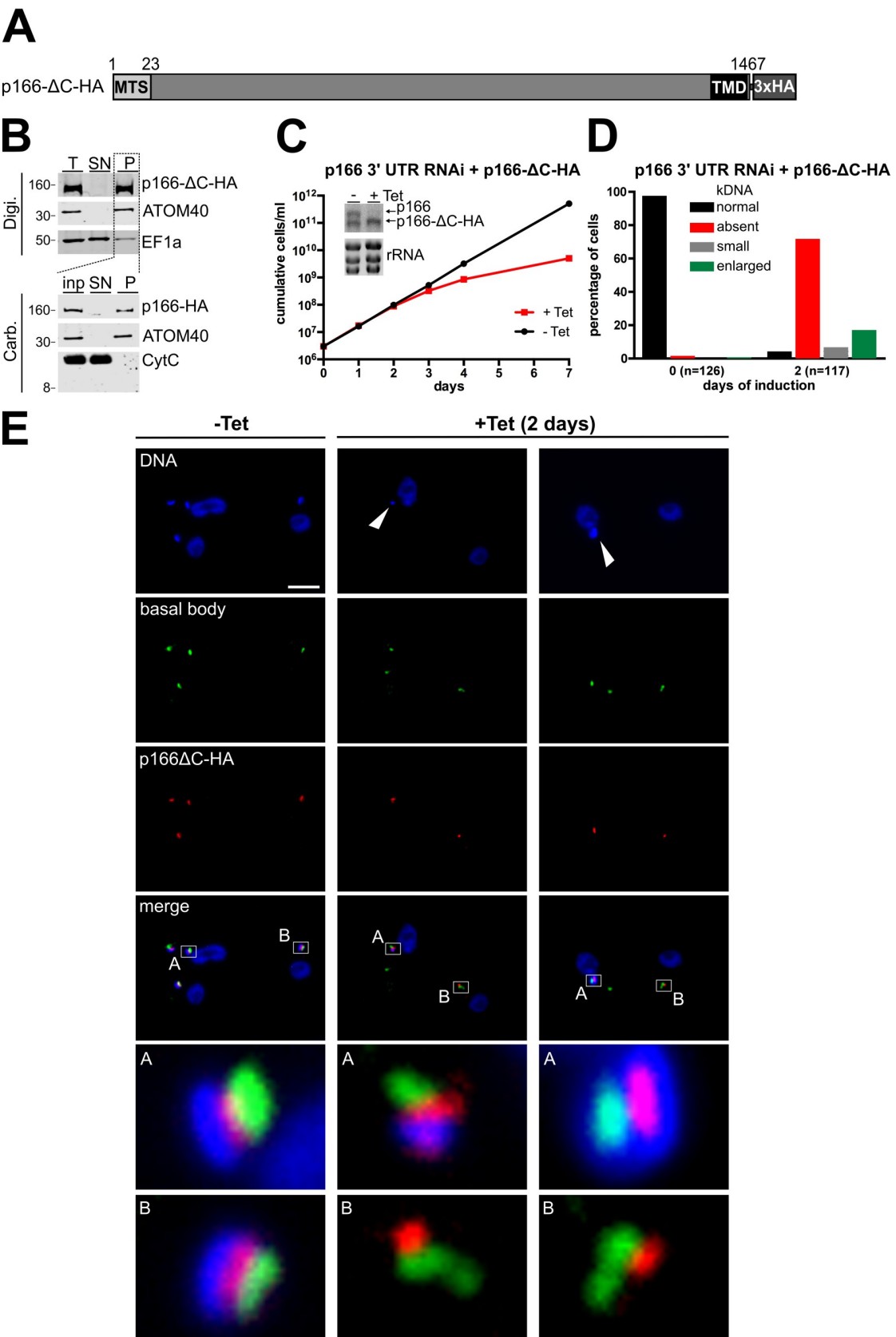

**Fig 3. The C-tail of p166 is essential for function. (A)** Schematic depiction (not to scale) of *in situ* tagged p166 lacking the 34 amino acid C-tail (p166-ΔC-HA). **(B)** Biochemical fractionation of cell extracts as described in legend 2B. **(C)** Growth analysis of cells uninduced (-Tet) or induced (+Tet) for RNAi against wild-type p166 with constitutive expression of p166-ΔC-HA as demonstrated by northern blot analysis. **(D)**. kDNA content of cells was analyzed as described in legend 1B. **(E)** Whole cells left uninduced (-Tet) or being induced for 2 days (+ Tet) were analyzed by immunofluorescence microscopy for DNA (stained with DAPI), the localization of basal bodies (using anti YL1/2) and p166-ΔC-HA. Enlarged merged pictures are shown and indicated. White arrowheads point at small kDNA (middle panel) and enlarged overreplicated kDNA (right panel). Scale bar, 5 μm.

the C-terminus, do not show a growth phenotype and their kDNA content stayed virtually unchanged over the whole course of the experiment (Fig 4B and 4C, right panels).

When whole cells were analyzed by immunofluorescence microscopy, both mini-versions of p166 were found distributed all over the mitochondrial membrane. The general mitochondrial staining is most likely caused by overexpression of the mini versions since the signals are not evenly distributed and local foci of higher signal strength could be observed some of which appear to be in close vicinity to the kDNA in the case of mini-p166-HA (Fig 4D). To better visualize a possible TAC localization we also analyzed Triton-X100 extracted cytoskeletons by immunofluorescence. Here, a clear colocalization of mini-p166-HA with basal bodies could be observed in more than 90% of the cells analyzed. In contrast, mini-p166-ΔC-HA colocalized with basal bodies in only 11% of the cells. (Fig 4E). Finally, isolated flagella were also analyzed. Here, mini-p166-HA colocalizes with basal bodies in almost 50% of the flagella investigated, whereas less than 3% of flagella showed a signal for mini-p166-ΔC-HA (Fig 4F).

We conclude that while both mini-versions are imported into mitochondria, only p166-HA is stably integrated into the growing TAC. p166-ΔC-HA on the other hand is mostly lost in isolated cytoskeleton and flagella possibly because its interaction with the TAC is Triton-X100 sensitive. Based on the findings that the TAC is assembled stepwise beginning from the basal body towards the kDNA and that addition of new TAC components needs the presence of stably integrated TAC components upstream [20] the presented results can best be explained in the following way: The C-tail of p166 reaches into the IMS and connects p166 to already integrated upstream TAC components resulting in a correct and stable TAC localization. Due to the polarized assembly of the TAC stable localization of mini-p166-HA with an intact C-terminus does not depend on the large matrix-exposed N-terminus of p166. Moreover, overexpression of mini-p166-HA likely is able to compete for localization with wild-type p166 and thus explains the slight dominant-negative effect on kDNA segregation and cell growth (Fig 4B, left panel). These effects are not seen for mini-p166-ΔC-HA.

To confirm the IMS localization of the C-terminus for full length p166 we did a protease protection assay using digitonin-extracted mitoplast fractions. Fig 4G shows that the C-terminal HA tag of p166 is removed by proteinase K and the same is the case for the OM protein ATOM69. However, the IM-localized mitochondrial carrier protein 5 (MCP5) that has multiple TMDs and the matrix marker mitochondrial heat shock protein (mHsp70) remain intact and are only digested after detergent solubilization of the IM. Since p166 contains a single TMD only, these results demonstrate that its C-terminus reaches into the IMS.

## The C-tail connects p166 to the OM subunit TAC60

Having established that the C-tail of p166 reaches into the IMS we wanted to directly test a putative interaction with well-studied OM TAC subunits, namely the beta-barrel proteins TAC40 and TAC42 as well as TAC60 which contains two TMDs (Fig 5A) [21, 22]. Since the functional full length p166 that is integrated into the TAC is not soluble (S2 Fig) we used cells expressing the mini-versions of p166 which can be fully solubilized by digitonin for pull down experiments. To that end cell lines were generated allowing simultaneous inducible

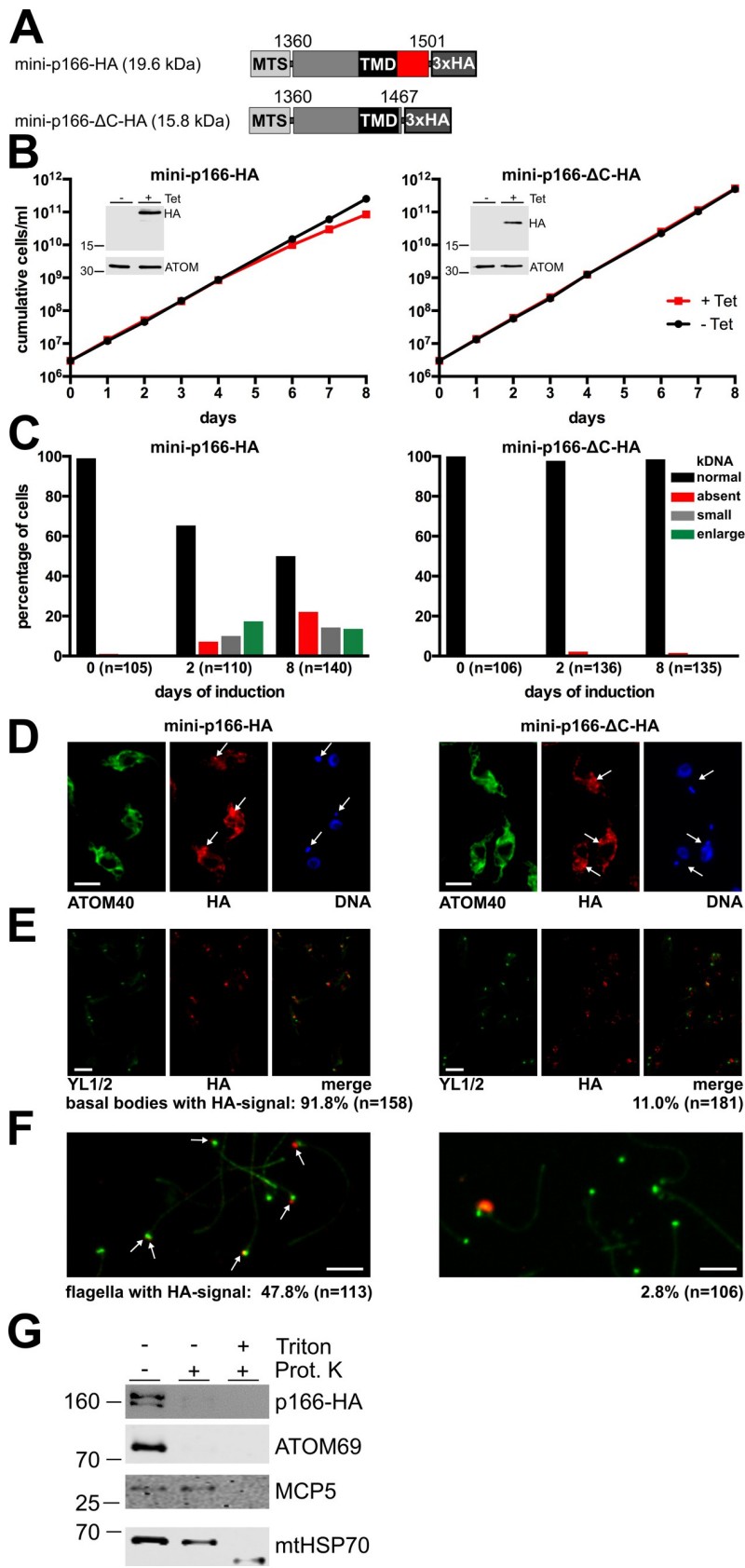

**Fig 4. The C-tail of p166 is located in the intermembrane space. (A)** Depiction of inducibly expressed N-terminal truncated p166 with (mini-p166-HA) or without the C-tail (mini-p166-ΔC-HA) and their calculated molecular weights. Numbers indicate amino acid residue positions based on wild-type p166. **(B and C)** Growth analysis of untreated cells (-Tet) and cells induced for expression (+Tet) of the indicated proteins and subsequent analysis of kDNA content. Inducible induction is shown by immunoblots probed for HA and ATOM40 as control for equal loading. **(D)** Immunofluorescence microscopy of cells induced for expression of the N-terminal truncated mini-p166-HA or mini-p166-ΔC-HA. anti-ATOM40 stains the mitochondrial membrane, DNA is visualized using DAPI. The arrows indicate foci with the highest signal strength derived from anti-HA. **(E)** Analysis of cytoskeletons for localization of basal bodies stained with anti-YL1/2 and HA-tagged mini-versions of p166. **(F)** Basal bodies (green) and *in situ* tagged truncated versions of p166-HA (red) were visualized by immunofluorescence microscopy in isolated flagella. Colocalization of both signals was analyzed in the indicated numbers of cells. Scale bar in D, E and F, 5 μm. **(G)** Mitoplasts were prepared from the cell line expressing in situ tagged full length p166 (p166-HA) using 0.05% of digitonin. Aliquots of the resulting pellet fraction were treated with proteinase K (10 μg/ml) and 0.5% (w/v) of Triton-X-100 as indicated. The atypical translocase of the outer membrane 69 (ATOM69), the IM protein mitochondrial carrier protein 5 (MCP5) and the matrix protein mitochondrial heat shock protein 70 (mtHsp70) served as controls.

overexpression of the c-myc tagged OM TAC subunits (TAC40, TAC42 and TAC60) together with the 3xHA tagged mini versions of p166. After 2 days of induction mitochondria of these cell lines were enriched by digitonin extraction and proteins solubilized from the pellets were subjected to anti c-myc immunoprecipitation. All three bait proteins TAC40, TAC42 and TAC60 were found to pull down mini-p166-HA, with TAC60 showing the highest enrichment. Importantly, the interaction with TAC60 was abolished in the pull-down with mini-p166-ΔC-HA highlighting the functional relevance of the C-tail (Fig 5B). TAC40, TAC42 and TAC60 were previously shown to form a stable subcomplex [22]. Hence, from our pulldown experiments with all three baits we cannot define which OM TAC subunit interacts directly with the C-tail of p166. However, based on its topology TAC60 is the most likely candidate. Previous mutational analyses revealed that its C-terminus is dispensable for function whereas a truncation of both the N- and the C-terminus leads to a correctly localized but non-functional TAC60 [22]. Fig 5C shows that both minimal versions of TAC60 are able to pull down p166-HA indicating that the interaction is at least partially mediated by the IMS loop of TAC60.

All immunoprecipitation discussed above were done under native conditions which results in the isolation of heteroligomeric protein complexes. It is therefore difficult to show whether any two proteins of the complex interact directly with each other. In order to circumvent that problem we co-expressed c-myc tagged TAC60ΔC283 and mini-p166-HA in *Saccharomyces cerevisiae*. Subsequently, yeast mitochondria were isolated, lysed and the soluble proteins were subjected to anti c-myc immunoprecipitation. The result shows that C-terminal truncated TAC60 efficiently pulls down mini-p166-HA but not yeast Tom40 (Fig 5D). Since TAC60 and mini-p166-HA are trypanosomatid-specific proteins that do not have orthologues in yeast, we conclude that p166 interacts directly with the IMS exposed loop of TAC60 (aa positions 142–237) without the need of any trypanosomal IMS protein. After having confirmed the interaction of TAC60 with the C-tail of mini-p166-HA we performed the reciprocal immunoprecipitation experiments with both mini-versions of p166 as bait proteins. In order to identify additional putative interaction partners we analyzed the isolated proteins by mass spectrometry combined with stable isotope labeling with amino acids in cell culture (SILAC) labeling. Our results show that immunoprecipitation of mini-p166-HA resulted in the enrichment of co-purified TAC40 and TAC60 confirming our previous results. In contrast, mini-p166-ΔC-HA did not interact with any known TAC subunit (S4 Fig and S1 Table). Other proteins that were higher enriched using the bait with the intact C-tail were four subunits of the ATOM complex (S4 Fig and S1 Table). The significance of this observation is presently unclear but it is consistent with the idea that a fraction of the ATOM complex might be associated with the TAC.

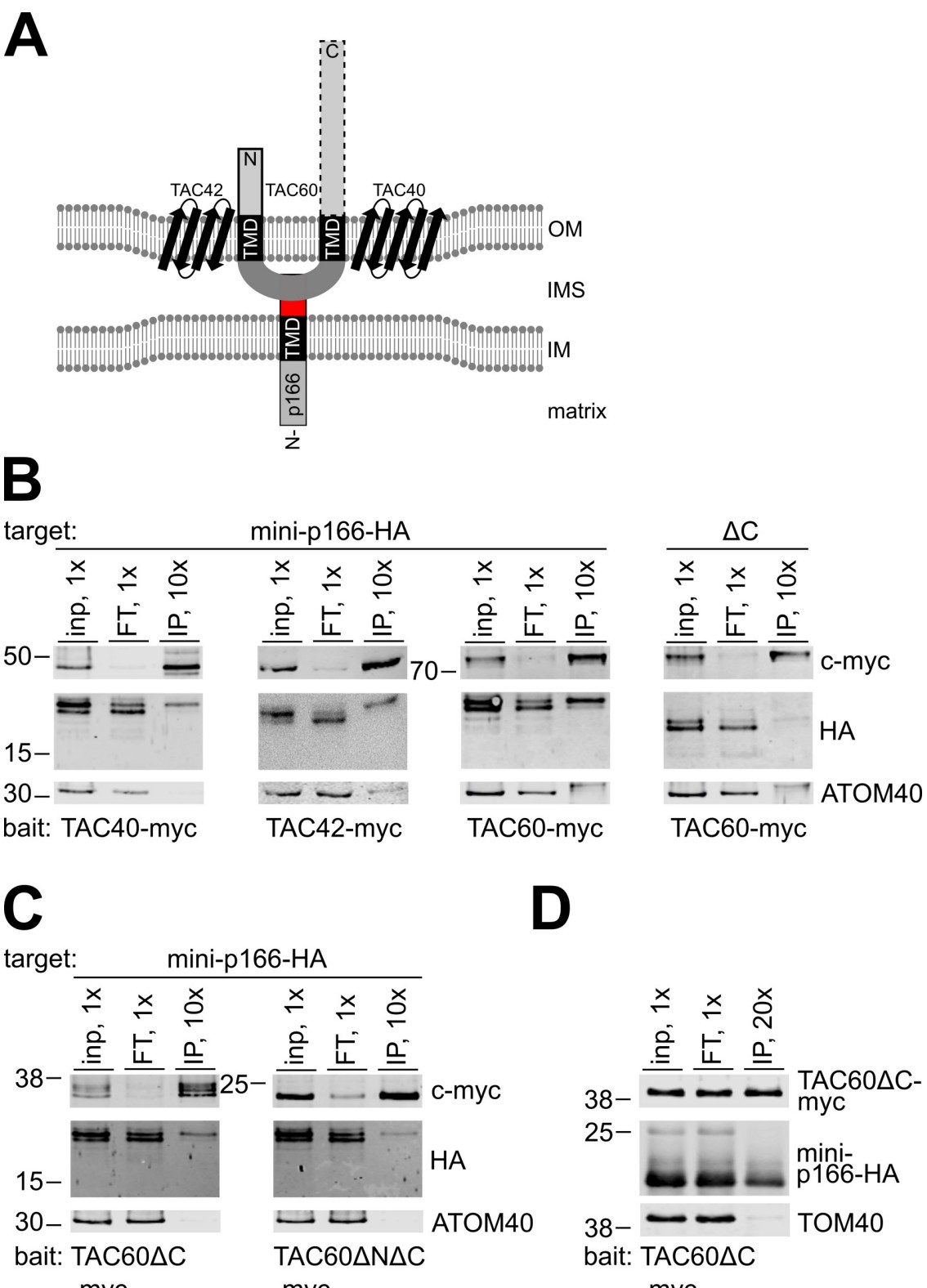

**Fig 5. The C-tail of p166 interacts with TAC components of the outer mitochondrial membrane. (A)** Model of possible interactions. TAC60 interacts with the beta barrel proteins TAC40 and TAC42 in the outer mitochondrial membrane (OM). It contains a region between two transmembrane domains (TMD) that is located in the intermembrane space (IMS) just like the C-tail

(red) of inner mitochondrial membrane (IM) anchored p166. Both the essential N-terminal domain (solid box) and the non-essential C-terminal domain (dotted box) are facing the cytoplasm. **(B and C)** Immunoprecipitation of indicated c-myc tagged TAC components expressed *in T. brucei* and analysis of copurification of N-terminal truncated tagged versions of p166 by immunoblot and detection of the indicated antigens. ATOM40 serves as a control for specificity **(D)** Immunoprecipitation of indicated proteins expressed in yeast. TAC60ΔC-myc serves as bait and TOM40 as control for specificity. Inp, input; FT, flow through; IP, immunoprecipitate. 1x, 10x, 20x indicate the relative cell equivalents loaded on the gels. Position of molecular weight marker bands with sizes in kDa are indicated.

## Localization of p166 is independent of the presence of C-tail or kDNA

The results presented so far indicate that the C-tail is necessary for integrating p166 into the TAC as it interacts with the IMS-exposed loop of the OM TAC subunit TAC60. In light of the current model of stepwise TAC assembly from the basal body towards the kDNA this raises the question how the non-functional p166-ΔC-HA is able to localize correctly without the possibility to interact with upstream TAC subunits (Fig 3E). Cells presented in Fig 3 express full length p166-ΔC-HA after RNAi mediated downregulation of wild-type p166. It is well possible that this knockdown is not complete and remaining amounts of wild-type p166 facilitate the localization of p166-ΔC-HA by homotypic interactions. An alternative explanation is the existence of so far unknown stabilizing factors in the IM that are kept in place by upstream TAC components and that interact with regions of p166 outside the C-tail.

To investigate this in more detail we switched to bloodstream form *T. brucei* as study objects. In particular, we chose the engineered cell line γL262P which grows normally in the complete absence of kDNA [27]. First, we knocked out one allele of p166 by replacement of the coding sequence with the gene for hygromycin resistance. Second, constructs for expression of *in situ* tagged full length p166-HA or p166-ΔC-HA were integrated in the remaining allele. As a consequence, both cell lines exclusively express the respective versions of tagged p166 (Figs 6A and S5). All four cell lines are viable and show no obvious difference in growth rates when cultured in identical standard conditions (Fig 6B). kDNA analysis confirmed that *in situ* tagged full length p166 is functional with the vast majority of cells exclusively expressing p166-HA showing normal amounts of kDNA. In contrast all cells expressing only p166-ΔC-HA lost their mitochondrial genome (Fig 6C). This again confirms that the C-tail is essential for p166 function in TAC mediated kDNA segregation.

When cytoskeletons were analyzed by immunofluorescence microscopy we noticed that both versions of p166 still locate close to the basal body (Fig 6D). This proves that non-functional p166-ΔC-HA can still localize correctly to the region of the TAC in the complete absence of either kDNA or wildtype p166. Thus, p166-ΔC-HA can localize to the kDNA-basal body region without having a C-tail that mediates the connection to upstream TAC components of the OM.

## Discussion

p166 was identified as the first subunit of the TAC in a pioneering study of the Englund lab using an RNAi library approach to screen for proteins involved in kDNA maintenance. Zhao et al. provided convincing evidence for p166 functioning in kDNA segregation rather than replication. In line with this, immunofluorescence analysis revealed that p166 is located between the basal body and the kDNA [19] (Fig 1C). However, it remained unclear where exactly within the three TAC regions p166 needs to be placed. The predicted presence of a TMD suggested it to be a membrane protein. The present study now provides biochemical evidence that p166 is indeed an integral membrane protein (Fig 2B). Furthermore, we show that the N-terminal 25 aa of p166 function as a MTS. This together with microscopic analyses in previous studies [20] let us conclude that p166 is localized in the IM and thus the first integral

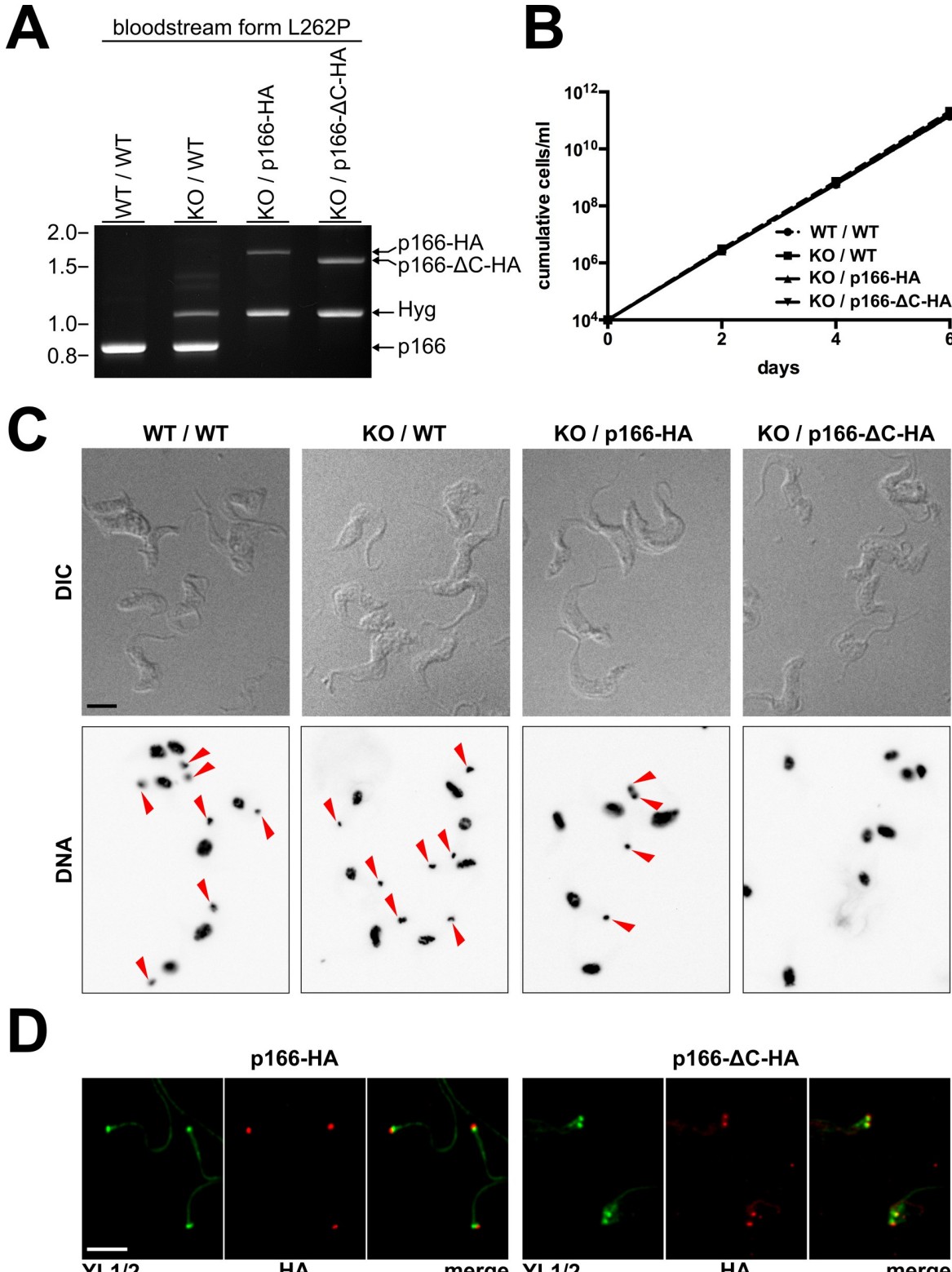

**Fig 6. C-terminal truncated p166 localizes correctly in the L262P bloodstream form cell line.** (**A**) Ethidiumbromide stained agarose gel after gelelectrophoresis of specific PCR products to analyze the p166 alleles of the indicated L262P bloodstream form cell line variants. Sizes of DNA molecular weight marker bands in kb are indicated on the left. (**B**) Growth curve analysis of cell lines harbouring indicated alleles. (**C**) Fluorescence microscopic analysis of the indicated L262P bloodstream form cell lines. Whole cells are visualized using DIC

and nuclei and kDNA are stained with DAPI. Red arrowheads indicate the presence of kDNA. **(D)** Anti-HA immunofluorescene microscopy on isolated cytoskeletons to visualize the localization of full length (p166-HA) or C-terminal truncated p166 (p166-ΔC-HA) in the KO / p166-HA and the KO / p166-ΔC-HA L262P bloodstream form cell lines. For comparison the basal bodies are stained with anti-YL1/2. Scale bars in C and D, 5 μm.

IM TAC subunit that has been identified. Our results show that p166 is anchored in the IM via its single C-terminal TMD thereby exposing the short C-tail to the IMS and the large N-terminal moiety into the mitochondrial matrix. This suggested topology of p166 is supported in several ways: (i) the predicted c-terminal TMD is necessary and sufficient for p166 to be integrated into the IM, (ii) the C-tail of p166 stably interacts with the IMS exposed loop of the OM subunit TAC60; (iii) this interaction is disrupted in mutant p166 lacking the C-tail (Fig 5) and (iv) a yeast two hybrid screen in a recent study revealed that the matrix TAC subunit TAC102 interacts with the N-terminus of p166 [28]. We therefore propose that p166 provides a direct connection of the OM TAC module with the matrix-localized TAC module. More specifically p166 bridges the distance between TAC60 in the OM and TAC102 in close vicinity of the kDNA [29]. Co-immunoprecipitation experiments in yeast expressing variants of TAC60 and p166 indicate that no additional—possibly IMS-located—trypanosomal factors are needed for this stable interaction. Fig 7A shows a model of the DM-ULF region of the TAC emphasizing the new results of the present study.

Presently p166 is the only known essential integral IM subunit of the TAC. Considering that p166 can bridge the gap from the OM to the kDNA, it is possible that p166 is the only IM TAC subunit required for TAC function. However, it is presently not possible to exclude that other as yet unknown IM TAC subunits might exist.

Intriguingly, we find 4 different integral membrane proteins in the OM (TAC40, TAC42, TAC60 and pATOM36) all of which were shown to be essential for TAC function [21–23]. Except for pATOM36 which has a second function in OM protein biogenesis [23], the phenotype of RNAi cell lines for the OM TAC subunits is essentially identical to what is observed for other TAC subunits such as p197 [30], p166 [19] and TAC102 [29]. This indicates that the function of TAC40, TAC42 and TAC60 is limited to the TAC raising the question why the TAC needs 4 subunits in the OM. Presently we cannot answer this question. However, there are two main scenarios. It is possible that the multiple TAC subunits in the OM could, at least in principle, be replaced by a single subunit. The seemingly unnecessary complexity of OM TAC region could in such a case be explained by a ratchet type mechanism termed constructive neutral evolution [31–33]. This evolutionary mechanism can increase the complexity of a structure without providing a selective advantage for the cell. Alternatively, having four OM TAC subunits may be the result of adaptive evolution. What this advantage for the cell might be is difficult to discern. Both the EZF on the outside as well as the ULF on the inside interact with the OM. It can therefore be speculated that the four OM TAC subunits, which includes two beta barrel proteins [21, 22], are required to form a highly defined platform in the OM that integrates the organization of the EZF with the organization of the ULF.

The orientation of p166 in the IM was determined by expression of mini-versions lacking a substantial part of the N-terminus. In immunofluorescence analyses of whole cells these versions were found spread in the entire IM (Fig 4D). This suggests that they are imported all over the mitochondrion by the mitochondrial protein translocases of trypanosomes, ATOM and TIM, which are also localized all over the OM and IM respectively [25, 34]. Subsequently p166 moves laterally in the IM towards its final destination and docks on already localized TAC subunits. A mutant mini-version lacking the C-tail cannot make this connection with

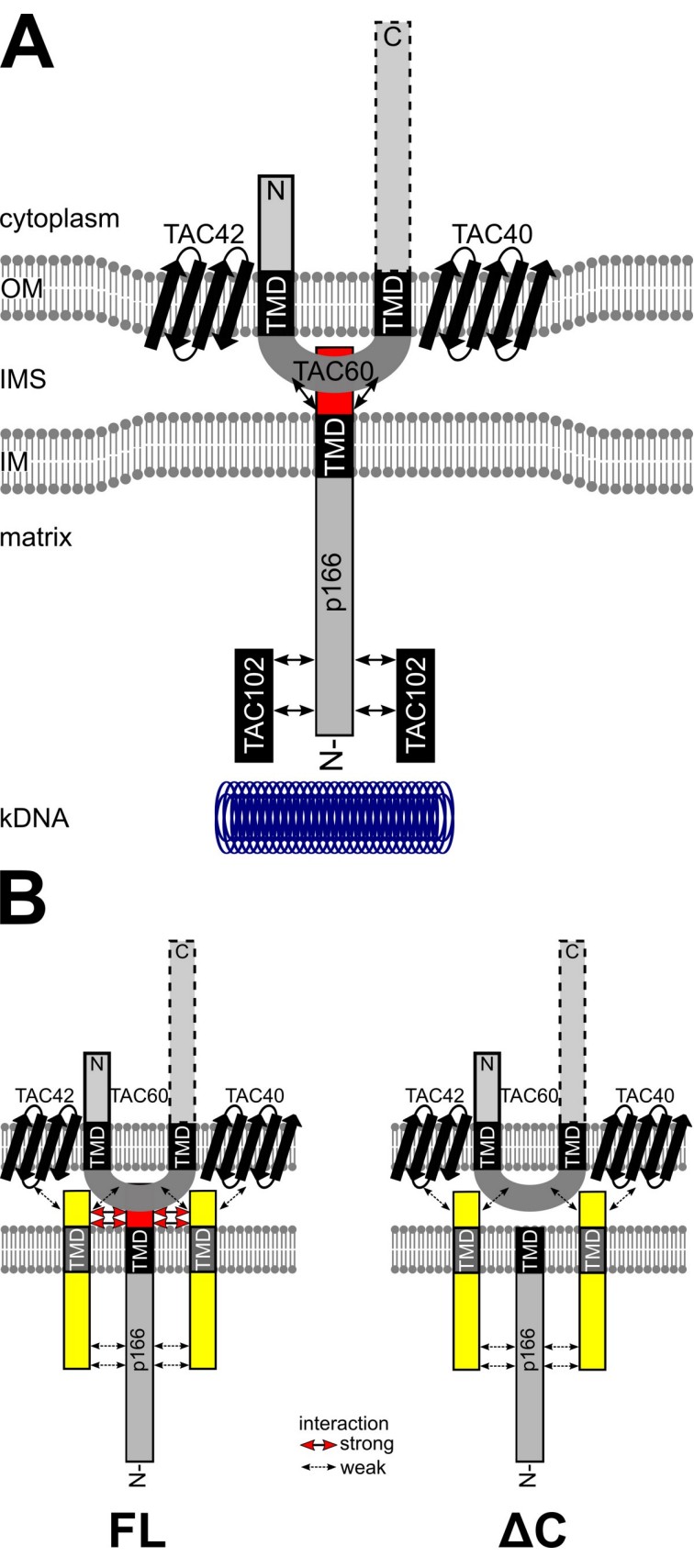

**Fig 7. p166 provides a bridge between OM and matrix located TAC components. (A)** Model of interactions of selected TAC components. The beta barrel proteins TAC40 and TAC42 form a stable subcomplex together with TAC60 in the outer membrane (OM). The C-terminal cytoplasmic domain of TAC60 is not essential for function (indicated by a dotted box) whereas the essential N-terminus (solid box) most likely provides the connection to cytoplasmic TAC subunits more proximal to the basal body. p166 is anchored in the inner membrane (IM) with a C-terminal transmembrane domain (TMD) exposing the short C-tail (red box) into the intermembrane space (IMS) where it interacts stably with the IMS exposed loop of TAC60. TAC102 interacts stably with the N-terminal part of p166 in the mitochondrial matrix and provides the connection to the kDNA disc. Protein-protein interactions are indicated by arrows. **(B)** Model of our working hypothesis on how non-functional C-terminally truncated p166 can be localized at the TAC. Full length (FL) p166 interacts via its intact C-tail (red box) with TAC60. So far unknown membrane anchored accessory factors (yellow) help to keep p166 in place. This is facilitated via strong interactions with the C-tail and weaker interactions with the N-terminus and OM subunits. p166 lacking the C-tail (ΔC) can still be locked in place by the aforementioned weak interactions. However, the lack of the C-tail disables a direct strong interaction with TAC60 and leads to a loss of kDNA in daughter cells due to insufficient mechanical stability of the mutant TAC.

OM located subunits and is–unlike the mini-version with a complete C-tail—not found concentrated in the TAC region highlighting again the functional importance of the C-tail.

In contrast to these results, non-functional p166-ΔC-HA with a complete N-terminal sequence does localize correctly in the absence of kDNA and a wild-type copy of p166 in both procyclic and bloodstream forms of the parasite (Figs 3E and 6D). A possible explanation for this observation is the existence of so far not identified factors that are able to stabilize the interaction of OM subunits with p166 in regions that are absent in the mini-versions of p166. These putative TAC associated proteins would be able to keep full length p166 at the TAC region irrespective of the presence or absence of its C-tail. The interaction between the postulated TAC-associated proteins and p166 would most likely be of low affinity, sufficiently stable to keep p166-ΔC at the TAC but not strong enough to withstand the mechanical force upon kDNA segregation which would explain the loss of kDNA in cells expressing C-terminally truncated p166. Our working model for such scenarios is presented in Fig 7B. We suggest that such stabilizing factors would be integral IM proteins whose IMS-domains could interact with OM TAC subunits. Their matrix-exposed domains on the other hand would bind to the N-terminal domain of p166 possibly with the predicted leucine zipper and/or other coiled-coiled motifs [19].

The identification of p166 as the very first TAC subunit is more than a decade ago [19]. Since then, intensive research activities led to the characterization of six TAC subunits meeting all criteria for TAC components and pATOM36 with another function in addition to kDNA segregation. These subunits reside in the cytoplasmic EZF (p197 and TAC65), the outer DM (pATOM36, TAC40, TAC42 and TAC60) and the ULF in the mitochondrial matrix (TAC102) [14]. p166 localizes to the inner DM and extends into the ULF and therefore represents the so far missing component allowing us to define subunits in all different compartments covered by the TAC. Our data suggest the presence of auxiliary proteins needed for assembly and stability of this essential structure. Hence, their identification and functional characterization will have to be the next level of TAC research in trypanosomes.

## Material and methods

### Transgenic cell lines

All procyclic cell lines are derivatives of *Trypanosoma brucei* 29–13 [35] grown at 27˚C in SDM-79 supplemented with 10% (v/v) fetal calf serum (FCS).

For tetracycline-inducible RNAi of p166 (Tb927.11.3290) cells were stably transfected with a NotI-linearized plasmid containing stem-loop sequences covering the 3' UTR positions 32 to 386.

*In situ* 3xHA-tagging of p166 was done by stable transfection of PCR products amplified using plasmids of the pMOtag series [36] with primers defining the sites of homologous recombination leading to expression of tagged full length p166 or a version lacking the last 34 amino acids comprising the C-tail.

A plasmid for inducible expression of POMP10-GFP [37] was used to generate the transfection vectors for studying the MTS of p166. In brief, POMP10 was excised with HindIII and AgeI and subsequently, the backbone was either re-ligated after Klenow treatment for expression of cytosolic GFP or filled with oligonucleotides hybridized to generate compatible ends encoding the MTS of either mtHSP60 or p166.

Genomic DNA of cells expressing *in situ* tagged p166 was prepared and used as template for the amplification of coding sequences of N-terminal truncated versions of p166. To ensure mitochondrial targeting of the overexpressed proteins the forward primer contained the sequence coding for the mitochondrial targeting sequence (MFRCVVRFGAKDIRF) of the mitochondrial matrix protein HSP60. PCR products were ligated into a derivative of pLEW100. The final plasmids were linearized with NotI before stable transfection.

The same constructs were stably transfected into cells capable of overexpression of triple c-myc tagged TAC40, TAC42 and TAC60 [21, 22].

Stably transfected bloodstream form *T. brucei* derive from a mutant cell line capable of surviving in the absence of kDNA [27]. These cells were cultured in HMI-9 containing 10% (v/v) FCS at 37˚C with 5% $CO_2$. A plasmid for knockout of p166 was generated as follows. The 5' flanking sequence of p166 comprising the nucleotide positions -513/-3 relative to the coding sequence and the 3' flank consisting of +4507/+5021 were amplified and ligated into XhoI/ HindIII and BamHI/SacI of pMOtag43M thereby flanking the hygromycin resistance gene. The resulting plasmid was digested with XhoI and SacI before electroporation. The resulting cell line was then further transfected with the described constructs for in situ tagging of the remaining p166 allele.

For co-expression of trypanosomal proteins in *Saccharomyces cerevisiae*, yeast cells (YPH499; MATa, ade2-101, his3-Δ200, leu2-Δ1, ura3-52, trp1-Δ63, lys2-801) were transformed with a derivative of pESC-HIS containing the coding sequences of C-terminal c-myc tagged TAC60ΔC283 and mini-p166-HA ligated into NotI/ClaI and BamHI/XhoI, respectively [38].

## Isolation of yeast mitochondria

Yeast cells were grown in SM-His medium (selective minimal medium, 0.67% [w/v] yeast nitrogen base without amino acids, 0.2% [w/v] SC amino acid mixture, 2% [w/v] glucose) at 30˚C. Cells were harvested by centrifugation and the yeast pellet was resuspended in YPGal medium (1% [w/v] yeast extract, 2% [w/v] bacto peptone, 2% [w/v] galactose) and grown for 18 hours at 30˚C. Cells were harvested in logarithmic growth phase and mitochondria were isolated by differential centrifugation [39]. Mitochondria were resuspended in SEM buffer (250 mM sucrose, 1 mM EDTA, 10 mM MOPS-KOH, pH 7.2), aliquoted, snap-frozen in liquid nitrogen and stored at –80˚C.

## Protein analysis

Digitonin extraction was performed to analyze the subcellular localization of proteins. $5x10^7$ cells were harvested, washed and resuspended in SoTE buffer (20 mM Tris HCl pH 7.5, 0.6 M sorbitol, 2 mM EDTA). An equal volume of SoTE containing 0.03% (w/v) digitonin was added for selective membrane permeabilization. After incubation on ice for 10 min the sample was

centrifuged for 5 min at 6800 g. Equal cell equivalents of whole cells, the supernatant containing cytoplasmic protein and the mitochondria-enrichded pellet were analyzed by immunoblot.

For alkaline carbonate extraction, mitochondria were enriched using the described digitonin fractionation and further separated into soluble and integral membrane proteins by incubation in 100 mM $Na_2CO_3$ pH 11.2 for 10 min on ice and a subsequent centrifugation for 10 min at 100'000g. Samples of the supernatant (soluble proteins) and pellet (membrane proteins) were subjected to immunoblot analysis.

For co-immunoprecipitation $1x10^8$ cells were used to generate a mitochondria-enriched pellet as described above. Mitochondria were solubilized by incubation in lysis buffer (20 mM Tris HCl pH 7.4, 100 mM NaCl, 0.1 mM EDTA, 10% glycerol) containing 1% digitonin and a protease inhibitor cocktail (Roche Complete, EDTA free) at three times the recommended concentration. After 15 min on ice, the sample was centrifuged for 15 min at 21'000g. The soluble supernatant was incubated with anti-c myc resin (Sigma) for 1 hour at 4˚C. Afterwards, the flow-through was collected, beads were washed 3 times with lysis buffer containing 0.1% digitonin and proteins were eluted by boiling the beads in SDS-PAGE sample loading buffer lacking β-mercaptoethanol. Fractions of the input, the flow-through and the entire eluted sample were precipitated using methanol and chloroform, resuspended in loading buffer containing β-mercaptoethanol and subjected to immunoblot analysis.

Co-immunoprecipitation of yeast expressed proteins was performed essentially in the same way using 250 μg isolated mitochondria lysed in the identical buffer as mentioned above.

## Immunofluorescence microscopy

For the analysis of proteins in whole cells, parasites were harvested, washed, resuspended in PBS and allowed to settle on slides for 10 min (procyclic forms) or 30 min (bloodstream forms). Cells were fixed using 4% paraformaldehyde and permeabilized with 0.2% Triton X-100.

For the analysis of cytoskeletons the order of procedures was reversed. Cells were first permeabilized for 1 min and subsequently fixed with paraformaldehyde.

Flagella were isolated as described by [40]. In brief, EDTA was added to the cultures at a final concentration of 5 mM before harvesting. Afterwards, cells were harvested by centrifugation, resuspended in extraction buffer (10mM $NaH_2PO_4$, 150mM NaCl, 1mM $MgCl_2$, pH 7.2) containing 0.5% Triton X-100 and incubated for 10 min on ice. After centrifugation and washing with extraction buffer cells were incubated in extraction buffer containing 1 mM $CaCl_2$ and incubated for 30 min on ice. Extracted flagella were pelleted by a 10 min spin, washed twice with PBS, resuspended in PBS and finally allowed to settle on glass slides. After fixing with paraformaldehyde samples were treated similar to whole cells or cytoskeletons.

Pictures were aquired using a DMI6000B microscope equipped with a DFC360 FX monochrome camera and LAS X software (Leica Microsystems). Images were further analyzed using Fiji software.

## SILAC-Immunoprecipitations and proteomics

Proteins for immunoprecipitations combined with SILAC proteomics were labeled in culture with light ($^{12}C_6/^{14}N_\chi$) and heavy ($^{13}C_6/^{15}N_\chi$) isotopes of arginine and lysine for 10 doubling times. Expression of the HA-tagged bait proteins was induced by addition of tetracycline for 2 days. Control cells were left uninduced. Induced and non-uninduced cells were mixed in a 1:1 ratio ($3.2x10^8$ cells in total). Lysates of mitochondria were produced as decribed above. The soluble supernatant was incubated with anti-HA affinity matrix (Roche) for 1 hour and all further steps were performed as describe above. Immunoprecipitation efficiency was confirmed

by immunoblot analysis and eluted proteins were analyzed by liquid chromatography-mass spectrometry (LC-MS) described below. SILAC experiments were done in four biological replicates including a label-switch.

Eluates from mini-p166-HA (n = 4) and mini-p166-ΔC-HA (n = 4) SILAC co-immunprecipitations were loaded onto SDS gels. Electrophoresis was performed until the proteins had migrated into the gel for approximately 1 cm. Protein-containing parts of the gel, visualised using colloidal Coomassie Blue, were excised, followed by reduction and alkylation of cysteine residues and tryptic in-gel digestion as described before [41]. Peptides were bound to in-house prepared 3-layer C18 stage tips, washed with 0.5% acetic acid and released with 80% acetonitrile (ACN)/0.5% acetic acid.

Tryptic peptides were analyzed by LC-MS using an Orbitrap QExactive Plus mass spectrometer (Thermo Fisher Scientific) connected to an UltiMate 3000 RSLCnano HPLC system (Thermo Fisher Scientific). Peptides were loaded and concentrated on a C18 precolumn (μPAC trapping column, Pharma Fluidics). For peptide separation, a C18 endcapped analytical column (50 cm μPAC column, Pharma Fluidics) and a binary solvent system consisting of 0.1% formic acid (solvent A) and 30% ACN/50% methanol/0.1% formic acid (solvent B) were used. Peptides were loaded and concentrated for 6 min at 5% solvent B, and the gradient for peptide elution was 5–22% B in 100 min, 22–42% B in 50 min, and 5 min at 80% B. MS parameters were as follows: $m/z$ 375–1700 for MS1 scans; 70,000 MS1 resolution (at $m/z$ 200); 3e6 MS1 automatic gain control (AGC) target; 60 ms MS1 maximum injection time; 28% normalised collision energy; 1e5 MS2 AGC; 120 ms MS2 maximum ion injection time; 3 $m/z$ isolation window; 35,000 MS2 resolution; 45 sec dynamic exclusion time; and +1 charged ions rejected.

Proteins were identified and SILAC ratios were determined using MaxQuant/Andromeda (version 2.0.2 [42, 43]). Database search was performed using the proteome of *T. brucei* TREU927 downloaded from the TriTrypDB (version 8.1, 11,067 entries, www.tritrypdb.org), the sequence of the respective min-p166-HA variant, mitochondrial proteins and a list of common contaminants provided by MaxQuant. Arg10 and Lys8 were specified as heavy SILAC labels, theoretical peptides were generated based on the sequence specificity of trypsin and allowing up to two missed cleavages. Mass tolerances were set to 20 ppm for the MaxQuant first search, to 4.5 ppm for the MaxQuant main search and to 0.5 Da for MS2 spectra. Carbamidomethylation of cysteine residues was considered as fixed modification and oxidation of methionine and acetylation of the protein N-terminus as variable modifications. The options 'requantify' and 'match between runs' were enabled. SILAC ratios (mini-p166-HA/control) were calculated based on at least two unique or razor peptides. The MaxQuant proteinGroups.txt file was parsed using Python with the pandas module. Protein groups for which ratios were calculated in less than 3 replicates were removed. Proteins significantly enriched with the respective mini-p166 construct compared to the control were identified using multivariable multiple regression (https://bioconductor.org/packages/release/bioc/html/limma.html) [44] with Benjamini-Hochberg correction of p-values.–$\log_{10}$ of the P values adjusted for multiple testing and $\log_{10}$ of the SILAC ratios were calculated for each protein group and visualised as volcano plots using matplotlib [45]. Results of protein identification and quantification are provided in the Supporting Information as S1 Table. The mass spectrometry proteomics data have been deposited to the ProteomeExchange Consortium via the PRIDE [46] partner repository with the dataset identifier PXD033042.

## Antibodies

All antibodies used in this study are described in detail [22].

## Supporting information

**S1 Fig. Biochemical fractionations reveal that p166 is an integral mitochondrial membrane protein and its single transmembrane domain is needed for membrane insertion. (A)** A cell line expressing p166 tagged in situ with HA at the C-terminus (p166-HA) was treated as follows. Cells were harvested, washed and the cell pellet was resuspended in SoTE buffer containing 0.015% digitonin (Dig.) to selectively permeabilize the cell membrane. After centrifugation tagged p166 is found in the mitochondria-enriched pellet fraction (P1) together with the mitochondrial proteins ATOM40 and Cytochrome C (CytC). In contrast, the cytoplasmic protein EF1a is found predominantly in the supernatant (SN1). Pellet P1 was further treated with 100 mM Na2CO3, pH 11.2 (Carb.). Subsequent ultracentrifugation results in a pellet fraction (P2) containing integral membrane proteins such as ATOM40 whereas the non-membrane associated cytochrome C is found in the supernatant S2. In a final step the pellet P2 was resupended with 1% Triton X-100 and centrifuged again. Analysis of the resulting supernatant (S3) and pellet (P3) shows that tagged p166 is found in the supernatant. Sensitivity to detergent proves that p166-HA pellets in carbonate extractions as true integral membrane rather than aggregated protein. **(B)** The same experiments were performed with lysates of a cell line expressing in situ HA-tagged p166 truncated to lack the predicted transmembrane domain (p166-ΔTMD-HA). In contrast to the full-length version, p166-ΔTMD-HA is found in the supernatant SN2 in alkaline carbonate extractions indicating that the transmembrane domain is essential for membrane insertion of p166.
(TIF)

**S2 Fig. Stably TAC-integrated p166 is not soluble.** Cells constitutively expressing in situ HA-tagged, RNAi-resistant p166 (p166-HA) were left uninduced or induced for RNAi to downregulate wild-type (WT) p166. Subsequently, mitochondria-enriched digitonin pellets were prepared and solubilized. After centrifugation, the soluble supernatant was subjected to BN-PAGE. Blots were probed with anti-HA to detect putative high molecular weight (HMW) complexes containing in situ tagged p166. A complex of around 440 kDa containing p166-HA can be detected if wild-type p166 is present. In contrast, in the absence of wild-type p166 when cells exclusively express the tagged version, the functional full length p166-HA is stably integrated in the TAC and becomes insoluble and therefore not detectable by BN-PAGE. Equal loading is demonstrated by reprobing the membrane with anti-ATOM40. Sizes of HMW marker bands in kDa are indicated on the left.
(TIF)

**S3 Fig. Overexpressed N-terminal truncated mini-versions of p166 are found in mitochondrial membranes. (A)** Schematic depiction (not to scale) of truncated p166-HA with or without the C-tail and their calculated molecular weight. The mitochondrial targeting sequence (MTS) of TbmtHSP60 was fused to the N-terminus. The amino acid positions of the remaining protein sequence of p166 are shown. **(B)** Digitonin (Digi.) and alkaline extraction using NaCO3 (Carb.) were performed using cells expressing mini-p166-HA containing (left panel) or lacking the C-tail (right panel). In both cases N-terminal truncated p166-HA is found in the mictochondria-enriched pellet after digitonin fractionation of cell extracts. Furthermore, if the digitonin pellet is subjected to carbonate extraction, both versions of mini-p166-HA are found in the pellet fraction indicating that they are integral membrane proteins. ATOM40 serves as marker for integral mitochondrial membrane proteins while EF1a and Cytochrome C (CytC) are markers for cytosolic and non-membraneous mitochondrial proteins, respectively.
(TIF)

**S4 Fig. The C-tail of mini-p166 is necessary to pull down OM TAC subunits TAC60 and TAC40.** Graphical visualization of quantitative mass spectrometry results obtained by SILA-C-IP using mini-p-166-HA (upper panel) and mini-p166- ΔC-HA (lower panel) as bait proteins. Both individual experiments were performed in 4 replicates. All proteins identified in at least 3 replicates are shown by single data points at which grey and red dots represent non-mitochondrial and mitochondrial proteins, respectively. Both bait proteins are indicated. Enrichment is shown on the x-axis in logarithmic scale. Significance of enrichment was tested using a one sided t-test (y-axis). Proteins enriched more than 2-fold are found right of the dashed line. Both data sets were searched for all known TAC and ATOM subunits. TAC60 and TAC40 were only pulled down when mini-p166 was used as a bait.
(TIF)

**S5 Fig. Generation of bloodstream form *T. brucei* expressing exclusively HA-tagged p166. (A)** Schematic depiction of alleles in wild-type (WT) cells and derivatives. First, a single knock-out (KO) of p166 was established by integration of a DNA fragment containing the gene for hygromycin (Hyg) resistance flanked by upstream and downstream homologous sequences (yellow). The resulting cell line was further transfected with PCR products whose integration leads to the exclusive expression of p166 with or without the C-tail and a blasticidin (Bla) resistance gene for selection. Oligonucleotides for PCR on genomic DNA of candidate clones and their site of hybridization are indicated by numbered arrows. The expected sizes of the different amplicons are indicated on the right. (B) Ethidiumbromide stained agarose gel showing the result of the aforementioned PCR test. Genomic DNA was isolated and used as template for amplification of specific regions in single PCR reactions containing all three different oligonucleotides. The final cell lines for exclusive expression of tagged p166 do not contain the wild-type allele anymore.
(TIF)

**S1 Table. Protein identification and quantification datasets for the SILAC-IPs using mini-p-166-HA and mini-p166- ΔC-HA as bait proteins.** The corresponding graph is shown in S4 Fig. The mass spectrometry proteomics data have been deposited to the ProteomeXchange Consortium via the PRIDE partner repository with the dataset identifier PXD033042.
(XLSX)

## Acknowledgments

We thank Silke Oeljeklaus for help with the data analysis and Elke Horn for technical support.

## Author Contributions

**Conceptualization:** Bernd Schimanski, Salome Aeschlimann, Philip Stettler, Sandro Käser, Bettina Warscheid, F.-Nora Vögtle, André Schneider.

**Funding acquisition:** Bettina Warscheid, F.-Nora Vögtle, André Schneider.

**Investigation:** Bernd Schimanski, Salome Aeschlimann, Philip Stettler, Sandro Käser, Maria Gomez-Fabra Gala, Julian Bender.

**Supervision:** Bettina Warscheid, F.-Nora Vögtle, André Schneider.

**Visualization:** Bernd Schimanski.

**Writing – original draft:** Bernd Schimanski, André Schneider.

**Writing – review & editing:** Bernd Schimanski, Salome Aeschlimann, Philip Stettler, Sandro Käser, Maria Gomez-Fabra Gala, Julian Bender, Bettina Warscheid, F.-Nora Vögtle, André Schneider.

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
