## [Decision Letter · Decision Letter 0]

18 Feb 2022

Dear Professor Schneider,

Thank you very much for submitting your manuscript "p166 links membrane and intramitochondrial modules of the trypanosomal tripartite attachment complex" for consideration at PLOS Pathogens. As with all papers reviewed by the journal, your manuscript was reviewed by members of the editorial board and by several independent reviewers. In light of the reviews (below this email), we would like to invite the resubmission of a significantly-revised version that takes into account the reviewers' comments.

One item that the authors need to address / clarify is the interpretation that p166 is the only inner membrane protein of the TAC. While p166 is convincingly demonstrated to be an essential component, is there enough evidence to support that p166 is the only essential inner membrane subunit and it is sufficient to bridge the distance between the OM and the kDNA.

Secondly, the authors need to provide more in depth analysis on the potential interactions with the p166 C-terminal domain.

The current findings and additional suggested analyses will be of great interest for others studying the TAC architecture.

We cannot make any decision about publication until we have seen the revised manuscript and your response to the reviewers' comments. Your revised manuscript is also likely to be sent to reviewers for further evaluation.

Sincerely,

Michele Klingbeil

Guest Editor

PLOS Pathogens

David Horn

Section Editor

PLOS Pathogens

Kasturi Haldar

Editor-in-Chief

PLOS Pathogens

orcid.org/0000-0001-5065-158X

Michael Malim

Editor-in-Chief

PLOS Pathogens

orcid.org/0000-0002-7699-2064

Reviewer's Responses to Questions

**Part I - Summary**

Reviewer #1: The mitochondrial genome of the parasite Trypanosoma brucei is connected to a complicated membrane structure, referred to as the tripartite attachment complex (TAC). The TAC spans through both mitochondrial membranes, thereby forming a bridge from the cytosol to the mitochondrial matrix. Several components of the TAC were described, however, the molecular architecture is still not entirely clear.

In this study, the authors characterize the TAC protein p166 and identify it as an inner membrane protein with a huge matrix domain and a short C-tail exposed to the intermembrane space (IMS). Depletion of p166 leads to the loss of mitochondrial genomes. The matrix protein TAC102 seems to be critical for mtDNA binding, in consistence with observations from the Ochsenreiter lab. Schimanski et al show that the short IMS domain of p166 makes the contact to the outer membrane protein TAC60, an interaction which even can be recapitulated in yeast, an organism which lacks a TAC equivalent.

This is an interesting study which identifies a central component in the mtDNA anchor complex of trypanosomes. The data are of high technical quality and well controlled. The topology of p166 is not unexpected given the N-terminal MTS and a C-terminal transmembrane domain. Both features were already recognized in the Zhao et al study (EMBO J, 2007) in which p166 was initially described. Nevertheless, the role of the C-terminal region remained unclear and it was speculated that the C-terminus of p166 interacts with the cytoskeleton directly. Thus, this paper now clarifies this critical point and serves as good basis to study the molecular structure and function of the TAC in more depth.

Reviewer #2: Reviewer’s comments PPATHOGENS-D-21-02538

In this manuscript, Schimanski et al., investigated the architecture of the tripartite attachment complex (TAC) that connects the flagellar basal body to the single mitochondrial genome known as kinetoplast (kDNA) through the double membrane of the mitochondrion in T. brucei. TAC is crucial for mitochondrial genome segregation during cell division.

Strength: Authors demonstrated that the C-terminal tail of p166, a previously identified TAC component, interact with the loop region of an outer membrane protein, TAC60, thus acts as a linker of the outer and inner module of the TAC. Although this interaction is not crucial for localization of p166 in the TAC, it is essential for k-DNA segregation. Most of the experiments are well done that provided important information regarding TAC.

Weaknesses: Some of the results are overinterpreted. Details comments are given below.

1. Previous studies showed that the full length p166 can be solubilized in 1% digitonin containing buffer and was analyzed by BN-PAGE (Hoffmann et al., PNAS 2018). Therefore, it is not clear why it is necessary to truncate a large N-terminal domain just to solubilize this protein.

2. The N-terminal truncated version of p166 distributed all over the mitochondria, which indicates that this region is critical for proper localization of this protein at the TAC. However, authors ignored this point till at the end of discussion. Instead, they mentioned (Page 7, 3rd paragraph) “mini-p166-HA with an intact C-terminus does not depend on the large matrix-exposed N-terminus of p166 for its localization”.

3. The legend of Fig. 4: This figure is not showing that the C-tail of p166 is in the IMS. Please modify the title of the figure legend

4. There is no evidence that the predicted TMD of p166 is responsible for membrane integration. Therefore, it is better to use the term ‘predicted’ throughout the manuscript.

5. Fig. 5B & C: there are multiple bands in the blots probed with anti-HA and anti-myc. Authors need to specify the non-specific bands

6. Zhou et al convincingly showed by immunogold EM that p166 is localized in the matrix side of the TAC (Zhao et al., EMBO J 2008). Therefore, the comment in the first paragraph of the discussion; “it remained unclear where exactly within the three TAC regions p166 needs to be placed” is not true. Authors need to modify this sentence.

7. P166 is most likely integrated in the IM exposing the smaller C-terminal tail in the IMS, however it is not experimentally verified. Which part of the protein is protected when mitoplast is treated with proteinase K?

8. Authors commented that P166 is the only IM protein of the TAC, which in contrast to the multiple components of the OM. This is true so far. However, other accessory proteins in the IM could be needed for TAC formation. Particularly when p166 is found in a 700 kDa complex (Hoffmann et al., PNAS 2018). What other proteins are present in this complex? Why p166 and TAC60 are not present in the same protein complex?

9. Some sentences in the abstract are over simplified. 1) “Surprisingly, non-functional p166 lacking the C-terminal 34 aa still localizes to the TAC region. This suggests the existence of non-essential TAC-associated proteins in the OM.” These non-essential TAC-associated proteins could be the filament-associated proteins or IM proteins, not only the OM-proteins. 2) “These proteins can loosely bind to non-functional p166 lacking the C-terminal 34 aa and keep it at the TAC but their binding would not be strong enough to withstand the mechanical force upon kDNA segregation.” This sentence requires modification. Due to lack of the connection between TAC60 and the C-tail of p166, kDNA was not segregated and stays attached to the old TAC, which indicates that without this interaction new TAC module is not properly formed and lose connection with the extended k-DNA. It is not clear what authors meant by mechanical force, which is undefined.

Reviewer #3: The manuscript by Schimansk et al. described a detailed analysis of p166, the first TAC component identified back in 2008. Based on current and previous studies, the authors proposed a model where p166 is a mitochondrial inner membrane (IM) protein, with the bulk N-terminal domain in the matrix and the C-terminal 34 amino acids interacting with TAC60 on the outer membrane (OM). While the C-terminal domain is not required for p166 localization to the TAC, it is essential for the normal functions of p166. The study identified p166 as an IM anchor that connects to the OM module to the kinetoplast. The interaction of p166 with other TAC components, particularly TAC60, is well established. The conclusion that p166 is the only essential IM subunit of TAC would require additional experimental support.

Reviewer #4: Pathogenic Kinetoplastids condense their mitochondrial genomes known as kinetoplast DNA or kDNA and physically connect them to the basal bodies of their flagella in order to obtain robust , reliable mitochondrial genome segregation. Subsequently, kDNA segregation occurs with and by the flagella during cell growth.

This manuscript addresses components in the kDNA basal body linkages and is a well written and well executed body of work. The authors identify p166 as the connective protein that traverses the inner mitochondrial membrane and places its C-terminus in the intra-membrane space. This allows p166 to interact with TAC60 and ultimately produces a system that links outer membrane proteins to the kDNA. They show that the C-ter of p166 is essential for proper TAC function and does not require kDNA for correct localisation. p166 is therefore the missing link in the kDNA-cytoplasm connection.

Reviewer #5: In this work, the authors describe the localization and interactions of p166, which was the first protein component of the TAC to be described. The data show that, as predicted, p166 is an integral membrane protein. They show that it has an N-terminal mitochondrial targeting signal and that it is part of the inner mitochondrial membrane. They show that the small C-terminus of p166 projects into the intermembrane space where it interacts with known TAC components. Interestingly, while the C-terminus of p166 appears to be critical for the function of the protein, it is mostly dispensable for its correct localization in the mitochondrial membrane, implying that the N-terminal region, found in the mitochondrial matrix, is involved in localization. Since a C-terminal truncated p166 still localizes to the correct region in the mitochondrion, there must be as yet undiscovered protein factors that interact with p166 and localize it to the TAC region.

This paper makes an important contribution in that, after so much activity in recent years to identify more TAC components and determine their localization, interactions, and order of assembly, it’s nice to see p166, the first TAC component, be integrated into this model, particularly in a way that opens up new questions for research in the field. It’s clear there are many aspects of this fascinating structure that are yet to be revealed, and this work stands as a very thorough investigation of this critical component. The experiments are convincing, careful, and well-presented.

**Part II – Major Issues: Key Experiments Required for Acceptance**

Reviewer #1: 1. The topology of p166 in the inner membrane is well documented in this study. However, the characterization of the matrix domain is very superficial and the statements to the model shown in Fig. 7B are a bit wishy-washy. The impressive length of this region might indicate complex interactions with other mitochondrial proteins as proposed here. The analysis of these interactions is certainly beyond the scope of this study. However, the authors should characterize the interactors of the C-terminal membrane region of p166 in more depth. To this end, they might employ their mini-p166-HA and �C constructs and subject the immunoprecipitates to mass spectrometry. Given the results shown in Fig. 4B, this should be a straight-forward strategy to characterize the molecular interactions in a more comprehensive and quantitative manner. Such an analysis would be of great interest for others studying the TAC architecture.

2. The authors identify the C-terminal 39 residues as a specific TAC60 binder. This p166-TAC60 interaction is one of the main conclusions of this study. However, it remains unclear how both proteins interact. According to the alphafold prediction, the IMS region of p166 is unstructured. The interaction assay in yeast shown in Fig. 4C is not very compelling as the p166-HA protein is almost exclusively found in the non-bound fraction. This study would even be stronger if the authors would have characterized the TAC60-p166 interaction in more depth, e.g. by a peptide spot assay or Biacore SPR analysis.

Reviewer #2: Additional experiments are needed to convincingly show that 1) the predicted TMD is responsible for membrane integration, and 2) the C-tail of p166 is located in the IMS.

Reviewer #3: 1. The MTS. The N-terminal MTS in p166 suggests its presence on IM. Figure 2C using MTS-GFP showed the mitochondrial localization nicely. To confirm the IM presence of the fusion construct and the topology of the protein as shown in Figures 5A and 7, MTS cleavage shall be tested using the MTS-GFP reporter. Presumably, the MTS is cleaved only if the N-terminal region of p166 is imported into the mitochondrial matrix.

2. The C-tail of p166 is not required for TAC localization. While this appeared to be the case for bloodstream form cells (Figure 6), p166-�C-HA seemed absent from the TAC in some procyclic cells (Figure 2E). In the +Tet samples, two cells with duplicated basal bodies were shown. p166-�C-HA appeared missing from the new basal bodies in both cells although it was present in cells with just one pair basal bodies but no kinetoplast. Could this be due to delayed assembly of the IM and other modules of the TAC in the absence of full length p166?

3. p166 is the only essential integral IM subunit of the TAC (page 9). Perhaps the authors meant to say that p166 is the only essential integral IM subunit identified so far? P166 is convincingly demonstrated to be an essential component, but I am not sure there is enough evidence to support that p166 is the only essential IM subunit and it is sufficient to bridge the distance between the OM and the kinetoplast (as stated in the abstract). To establish sufficiency, think a (semi)-purified system showing that purified p166 alone can link up OM and kintoplast complexes would be required. The current study cannot rule out the presence of other IM or intermembrane subunits required for this bridging, and the data and the model depicted in Figure 7 in fact predicted the involvement of other IM subunit(s).

Reviewer #4: These results provide important answers to questions about the formation and components of the TAC, and clearly provides new insights, but there are some unanswered questions. I think that this paper is pretty close to be ready for publication in PLoS Path when the corrections below have been adequately addressed.

Major comments

1. Figures 1B, 3D, and 4C, there is no visible statistical robustness in these data (error bars) and this, at least, must be addressed.

2. Immunoprecipitation using mini-p166-HA as bait would provide some clues as to whether the hypothesis in Fig. 7 is true. What are the proteins identified by IP of mini-p166-HA?

Reviewer #5: No major issues.

**Part III – Minor Issues: Editorial and Data Presentation Modifications**

Reviewer #1: 3. Fig. 1. Labeling is absent in panel B. Please copy the description from A also to the B panel

4. Fig. 5, legend. (D) should be in bold

Reviewer #2: Minor points:

- Page 5 line 11; Remove ‘exclusively’. As mentioned by the authors in later section that RNAi may not remove 100% of the wildtype copy.

- Page 5, line 29; replace ‘distant’ by ‘distal’

Reviewer #3: Page 7, “less then 3%” shall be “less than 3%”.

Reviewer #4: Minor comments

p166 does not require kDNA to target to the TAC. Where does it enter the mitochondrion and how does it find the TAC? The authors touch upon this in the discussion but is there solid proof that ATOM and/or TIM translocate p166? These are fundamental questions that would help understand how the TAC is built.

Figure 1B right panel. After knocking down endogenous p166 the HA tagged version is able to compensate but there are mutant phenotypes present. Please explain the reason for their production.

Figure 1B is not clear. Please label the graphs as WT and p166-HA or something similar and please label all graphs.

Page 4 last sentence, I am not sure what is meant by “After 2 days of induction mitochondria were enriched by digitonin extraction“. I think that there is a word(s) missing between “mitochondria” and “were”.

Figure 5 A, the legend does not match the image. There is no dotted box (C-terminal domain) in my image. Also, it may be obvious to some but “inp, FT and IP” need to be described in the legend. Please explain what the 1X and 10 mean on these figures. For 5D, TAC60deltaC-myc is the bait so Please indicate on the figure.

Figure 5C right panel. Why does the amount of mini-p166-HA drop in the delta N+C-ter IP bait versus the delta C-ter alone bait if the IMS region of TAC 60 binds to mini-p166-HA ?

Page 7. The sentence; “p166-DC-HA on the other hand is mostly lost in isolated cytoskeleton and flagella possibly because its interaction with the TAC is Triton-X100 sensitive”, does not quite make sense. Should it be more like; p166-DC-HA on the other hand is mostly lost in isolated cytoskeleton and flagella preparations possibly because its interaction with the TAC is Triton-X100 sensitive. Or, p166-DC-HA on the other hand is mostly absent when observing isolated cytoskeletons and flagella possibly because its interaction with the TAC is Triton-X100 sensitive”.

Reviewer #5: The authors state in the abstract, and explore further in the discussion, the idea that p166 is the only TAC subunit in inner membrane. Do the authors mean the only one identified so far or do they think, based on the fairly exhaustive screens that have been performed by this group and others, that all of the core TAC components have been found and that p166 really is the only one in the IM? In contrast to this, the authors also appear to argue that p166, since it localizes to the TAC region in the absence of kDNA and its C-terminal domain, relies on other factors to correctly position it in the differentiated membrane. They even propose in their model that these accessory proteins are probably integral proteins in the IM, and highlight this as an intriguing avenue for future investigation. Wouldn’t these proteins also count as IM TAC components? Do the authors distinguish between “core” and “accessory” components of TAC? This may just be a semantics issue but I think the manuscript would benefit from additional clarity on how the authors are defining true components of TAC versus TAC-interacting/assembly proteins.

The abstract states that non-essential TAC-associated proteins in the OM must retain the p166 lacking its C-terminal domain in the TAC region. Perhaps they meant TAC-associated proteins in the matrix or IM perhaps? And how do they know these proteins won’t be essential since they may be necessary for the correct localization of p166?

Along these lines, the authors point out the interaction between the N-terminal portion of p166 and TAC102. Do they think TAC102 is insufficient to provide the “weak interactions” necessary to retain p166 in the TAC region? In other words, why is it more likely to be an integral membrane protein as opposed to a kDNA binding protein? Hoffmann et al., 2018 seemed to find that p166 localization was disrupted during TAC102 knockdown by RNAi, although some remained. Residual localization of p166 could be is due to incomplete knockdown of TAC102?

Bottom of page 6, first paragraph under the heading “The C-tail of p166 reaches into the IMS”. I am convinced by the data in this paper that p166 is in the IM with the topology suggested (N-term in matrix) but is the distance between the two membranes in the differentiated portion really a compelling addition to this argument? Perhaps this could be omitted.

PDF version—arrows in figure 4 left panels seem to be partially obscuring areas of enrichment. Could they be shifted over?

Fig 5—maybe it’s just the PDF version again but I don’t see a dotted box for the c-term of TAC60.

Fig 5 IPs—why are there two bands for p166 and why does only one appear to be pulled down? Do the authors mention the ATOM40 negative control in the text? Information on bait/target should be indicated on Fig 5D or included in the figure legend. Should also include the information on panel or legend that TOM40 a yeast protein that is serving as a control. In Fig. 5D there are again two bands. I assume p166 is the lower band but why does it appear to run so much smaller than the mini-p166-HA shown in 5B?

In the text for Fig 6 (top of page 9), it was a little unclear what the authors meant by “four cell lines”. Perhaps the diagrams of the p166 locus found in supplemental Fig S3 showing where their primers bind could be moved to the main figure. That would help with interpretation of the gel as well. Alternatively, the text could be altered to make it clear what the four cell lines are.

PLOS authors have the option to publish the peer review history of their article (what does this mean?). If published, this will include your full peer review and any attached files.

Reviewer #1: **Yes: **Johannes M Herrmann

Reviewer #2: No

Reviewer #3: No

Reviewer #4: No

Reviewer #5: No
---

## [Editor Report · Decision Letter 1]

3 May 2022

Dear Professor  Schneider,

We are pleased to inform you that your manuscript 'p166 links membrane and intramitochondrial modules of the trypanosomal tripartite attachment complex' has been provisionally accepted for publication in PLOS Pathogens.

We appreciate your responses to reviewers' comments, the additional experiments 
and new data, and clarification of several other points raised by reviewers to provide a well-executed body of work.  All these items have been addressed to our satisfaction. 

Best regards,

Michele Klingbeil

Guest Editor

PLOS Pathogens

David Horn

Section Editor

PLOS Pathogens

Kasturi Haldar

Editor-in-Chief

PLOS Pathogens

orcid.org/0000-0001-5065-158X

Michael Malim

Editor-in-Chief

PLOS Pathogens

orcid.org/0000-0002-7699-2064
---

## [Editor Report · Acceptance letter]

10 Jun 2022

Dear Prof Schneider,

We are delighted to inform you that your manuscript, "p166 links membrane and intramitochondrial modules of the trypanosomal tripartite attachment complex," has been formally accepted for publication in PLOS Pathogens.

Best regards,

Kasturi Haldar

Editor-in-Chief

PLOS Pathogens

orcid.org/0000-0001-5065-158X

Michael Malim

Editor-in-Chief

PLOS Pathogens

orcid.org/0000-0002-7699-2064